

# Vestibulospinal reflexes elicited with a tone burst method are dependent on spatial orientation

Vincent Jecko[1,2], Léa Garcia[2], Emilie Doat[2], Vincent Leconte[2], Dominique Liguoro[1], Jean-René Cazalets[2] and Etienne Guillaud[2]

[1] Department of Neurosurgery A, University Hospital of Bordeaux, Bordeaux, France
[2] Univ. Bordeaux, CNRS, INCIA, UMR 5287, Bordeaux, France

## ABSTRACT

Balance involves several sensory modalities including vision, proprioception and the vestibular system. This study aims to investigate vestibulospinal activation elicited by tone burst stimulation in various muscles and how head position influences these responses. We recorded electromyogram (EMG) responses in different muscles (sternocleidomastoid-SCM, cervical erector spinae-ES-C, lumbar erector spinae-ES-L, gastrocnemius-G, and tibialis anterior-TA) of healthy participants using tone burst stimulation applied to the vestibular system. We also evaluated how head position affected the responses. Tone burst stimulation elicited reproducible vestibulospinal reflexes in the SCM and ES-C muscles, while responses in the distal muscles (ES-L, G, and TA) were less consistent among participants. The magnitude and polarity of the responses were influenced by the head position relative to the cervical spine. When the head was rotated or tilted, the polarity of the vestibulospinal responses changed, indicating the integration of vestibular and proprioceptive inputs in generating these reflexes. Overall, our study provides valuable insights into the complexity of vestibulospinal reflexes and their modulation by head position. However, the high variability in responses in some muscles limits their clinical application. These findings may have implications for future research in understanding vestibular function and its role in posture and movement control.

## INTRODUCTION

Balance disturbances, which can be caused by impairments in the vestibular system, including the inner ear receptor systems or alterations in vestibular pathways, may contribute to static spine disorders, movement difficulties, or low back pain (*Ruhe, Fejer & Walker, 2011*). The tone burst-evoked vestibulocolic reflex technique is a widely used and cost-effective method for clinical investigation of vestibular function. This technique, based on sound stimulation, has become the gold standard in clinical settings (*Welgampola & Colebatch, 2005*). Compared to methods like galvanic vestibular stimulation (GVS), the click-evoked technique is less invasive and more suitable for studying vestibular function in vulnerable clinical populations. Previous research has suggested that this technique

Corresponding author
Etienne Guillaud, etienne.guillaud@u-bordeaux.fr

specifically activates the saccule (*Young, Fernández & Goldberg, 1977*; *Zhu et al., 2014*; *Curthoys et al., 2016*).

The investigation of motor components involved in vestibular reflex pathways relies on the recording of target muscle activity. Electromyogram (EMG) recordings provide a means to assess vestibular evoked myogenic potentials (VEMPs), which are specific manifestations of vestibular reflexes (*Curthoys et al., 2018*). Initial research primarily focused on the vestibulocollic reflex, which involves the sternocleidomastoid (SCM) muscles and VEMPs recorded from the SCM muscles exhibit two waves of opposite polarity with distinct onset latencies of approximately 13 ms and 23 ms (*Colebatch & Halmagyi, 1992*). However, other studies have also described vestibulospinal reflexes induced by tone bursts in muscles such as the triceps brachii, triceps surae, extrinsic ocular musculature, and gastrocnemius muscle (*Rudisill & Hain, 2008*; *Cherchi et al., 2009*). In contrast to the investigations of vestibulospinal reflexes in appendicular muscles, the vestibulospinal reflex in axial muscles has primarily been explored using GVS in studies that measured trunk deviation resulting from postural compensation (*Maaswinkel, Veeger & Dieen, 2014*) or utilized electromyography to evaluate muscle activity (*Iles et al., 2007*; *Guillaud et al., 2020a*). These studies have revealed that different vestibulospinal pathways are involved in the control of trunk and leg muscles (*Guillaud et al., 2020a*). Furthermore, GVS experiments have provided evidence for the modulation of vestibular responses by visual, proprioceptive and somatosensory information (*Brandt, 1998*; *Iles et al., 2007*). Among these sensory inputs, proprioceptive information originating from the neck plays a crucial role in determining the spatial orientation of vestibulospinal reflexes (*Pastor, Day & Marsden, 1993*; *Fitzpatrick & Day, 2004*). By providing essential information on the position of the head in relation to the trunk, they help generate appropriate motor responses to displacements detected by the vestibular system. This integration of proprioceptive and vestibular inputs contributes to the precise control of balance and spatial orientation in various motor tasks. However, there is currently a lack of reports on such interaction when the vestibulospinal reflexes are elicited under acoustic stimulation.

The first aim of the current study was to investigate various vestibulospinal pathways involving axial and appendicular muscles using the tone burst stimulation method in the same experimental conditions. Initially, we compared the tone burst responses evoked in the sternocleidomastoid (SCM) muscles, which served as a reference, with those elicited in axial muscles such as the erector spinae (ES), and appendicular muscles such as the gastrocnemius (G) and tibialis anterior (TA). The secondary aim of this study, considering the crucial role of the vestibular system in spatial orientation, was to examine whether the integration of vestibulospinal responses evoked by the tone burst method was influenced by the postural context, as previously observed with GVS (*Forbes et al., 2015*). Our findings indicated that the position of the head had a significant impact on the elicited tone burst responses, suggesting a potential dependency on spatial factors.

## MATERIAL AND METHODS

In a first recording session (referred to as rs1), sixteen healthy participants were recruited (11 males, 5 females, aged $24.7 \pm 6$ years), and in the second recording session (referred

to as rs2), twenty healthy participants were recruited (12 males, eight females, aged 23 ± 2 years). They were recruited from among students at Bordeaux University. None of them reported a history of hearing problems, neurological impairments, or vestibular pathologies. Participants provided written informed consent. The experiments conducted in this study were approved by the ethical research committee (French Comité de Protection des Personnes Sud-Méditerranée I, ID RCB 2019-A02985-52).

Sound stimulations were administered unilaterally using TDH-49 Telephonics® supra-aural headphones to stimulate the saccules and elicit vestibular responses. Each test trial lasted for 50 s, during which the participants were exposed to 260 tone burst stimulations, resulting in an average rate of 5.2 stimulations per second. For each posture, participants underwent four similar 50-second trials, which were analyzed offline. Each trial was separated by a rest period of 1 min. The stimulation itself had a duration of 5 ms and followed a trapezoidal sinusoid shape with an ascent/descent time of 1 ms, a plateau of 3 ms, and an oscillation frequency of 500 Hz ± 5–7.

The stimulations were created using MATLAB software (MathWorks, Natick, MA, USA) and generated by a National Instruments analog card (NI, Austin, TX, USA) USB-6259 at a sampling rate of 20 kHz. The generated signals were then transmitted to a Luxman A-312 audio amplifier (Luxman Corp, Kanagawa, Japan). The TDH-49 earphones were directly connected to the amplifier output and received a 5V signal amplitude at peak. In terms of acoustic reference, this corresponds to a sound intensity of 100 dB. Simultaneously, white noise was delivered into the "non-stimulated" ear. Additionally, a copy of the sound signal (sampled at 10 kHz) was measured using a second analog card, Heka ITC-18 (Heka, Lambrecht, Germany), in parallel with electromyographic recordings.

The sound stimulus characteristics described earlier were determined to be optimal for eliciting VEMP responses with maximum amplitude while using the lowest possible stimulus intensity (*Papathanasiou et al., 2014*; *Janky & Shepard, 2009*). These characteristics were chosen to ensure reliable and consistent responses. The replication rate of the stimulus at 5.2 Hz, ensured a desynchronization between the frequency of exposure to the stimulus and the electrical current frequency (50 Hz). Myoelectric signals were recorded using surface EMG electrodes. The activity of the various muscle was monitored bilaterally, allowing for the analysis of the responses associated with different vestibular pathways.

During the first recording session (rs1), the following muscles were bilaterally recorded: sternocleidomastoid (SCM, neck muscle), lumbar erector spinae (ES-L, trunk muscle at the level of the third lumbar vertebra), medial gastrocnemius (G), and tibialis anterior (TA, leg muscles). In the second recording session (rs2), sternocleidomastoid and cervical erector spinae (ES-C, neck muscle at the level of the third cervical vertebra) were recorded bilaterally. To ensure good conductivity and signal recording, the skin was cleaned with 70% alcohol. Surface EMG electrodes were then applied to the contracted muscles, with the dipole orientation parallel to the muscle fibers. The EMG signals were amplified using a BTS TeleEMG system (BTS SPA, Milan, Italy) and digitized by the ITC18 analog card at a sampling rate of 10 kHz, along with the copy of the sound stimulation. For each muscle, the active electrode was positioned above the motor point, while the reference electrode

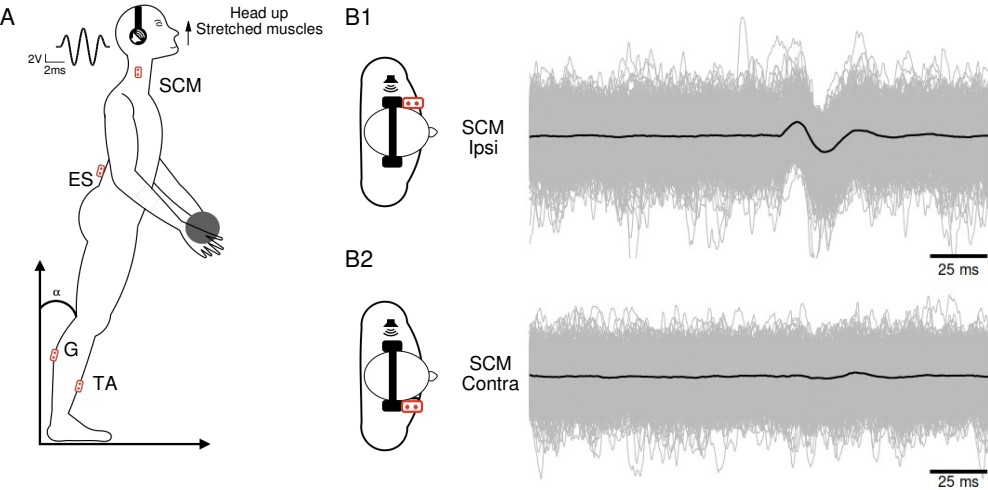

**Figure 1** **Subject's position and electromyographic tracings.** (A) Subject's position during the head-up condition, along with the locations of the recording electrodes. The waveform corresponds to the 5ms Gaussian stimulus. (B) Superposition of 1,040 electromyographic tracings (in gray) collected from a participant's ipsilateral (B1) and contralateral (B2) sternocleidomastoid following sound stimulations. The black curves represent the averages of the participant's 1,040 tracings in each condition.

was placed on the adjacent bone. A single ground electrode was located on the forehead. In the neck, the active electrode for the SCM muscles was positioned in the upper third of the muscles, with the references placed on the medial part of the clavicles. For the trunk muscles, the electrodes were placed at the level of the 3rd lumbar vertebrae, approximately 4.5 cm from the midline (*de Sèze & Cazalets, 2008*). In the lower limb, the active and reference electrodes were positioned on the G muscles and the right TA muscle.

To enhance the visibility of muscle responses, the recordings were conducted while the muscles were actively contracted. Participants stood with their feet naturally apart and their eyes closed. They assumed a position that allowed for a tonic contraction of the specific muscles of interest (Fig. 1A), which varied depending on the condition : (1) for the SCM muscle, the head was turned to the right, left, or raised (with the chin up; Fig. 1A) ; (2) for the ES muscle, the participants arched their back by approximately 7° between the T11 and L5 vertebrae. Additionally, the muscular tension was increased by holding a weight of 1 kg (Fig. 1A) ; (3) for the G and TA muscles, the participants shifted their center of gravity forward by bending approximately 15° .

Three different head positions were employed to investigate the impact of spatial orientation on vestibular reflexes. These positions included: (1) head up (HU) condition: the head was slightly tilted backwards; (2) lateral rotation: the head was rotated either to the left or right; (3) control condition: the head remained in a straight and neutral position without any tilting or rotation. During the stimulation phase, participants were instructed to keep their head still to prevent any unintentional displacements that could generate uncontrolled vestibular inputs. For each participant, six conditions were therefore defined in recording session 1 (rs1), which consisted of monaural stimulation (either right or left)

in each of the three head position conditions: Head Up (HU), and Head Rotated (either to the right or left). In recording session 2 (rs2), the head-up condition was not tested. Each condition included four trials, resulting in a total of 24 trials per participant. To mitigate any potential bias related to participant fatigue or habituation, the conditions were presented in a random order during the experiment.

## Signal analysis and processing

For data collection we used a custom-built toolbox (Oyapock) developed with MATLAB (*Guillaud et al., 2020a*). The collected EMG signals underwent several processing steps. Firstly, a Butterworth filter with a high-pass cutoff of 10 Hz and a fourth-order filter was applied to remove signal drift and movement artifacts. When necessary, the parts of ES muscle EMG affected by cardiac electrical activity were eliminated. To account for the large inter-individual variability in EMG amplitudes, normalization was performed. For each trial of a participant, the raw EMG sequences were normalized to the median of its absolute value, and expressed as arbitrary units (a.u.). Each EMG sequence comprised a 100 ms prestimulation recording period and a 100 ms post-stimulation recording segment. For each participant and condition, the 1,040 records were synchronized based on the stimulation onset and then averaged (Fig. 1B). The resulting average traces (depicted as black traces in Fig. 1B) were analyzed to identify muscle responses. An algorithm was employed to detect a response when the signal amplitude exceeded three times the standard deviation. We calculated the ratio of participants who exhibited a response magnitude exceeding three times SD of the pre-stimulus control period for each muscle and in each condition. Once a response was identified, the positive and negative peaks were visually determined, and an algorithm detected the onset and end times of these peaks. By comparing the positive and negative peak latencies, the response waveform (positive then negative or vice versa) was characterized.

## Statistical analysis

The stimulations (S) and head orientations (H) were classified as ipsilateral (SI and HI) or contralateral (SC and HC) to each recorded muscle in order to analyze the measurements of both right and left muscles of the participants. These responses were symmetrical. To compare the averages calculated in each condition, repeated measures ANOVAs were conducted. A series of ANOVAs was performed to compare the amplitude of each muscle response. This analysis used a two-factor design with three head positions (HU, HI, HC) and two stimulation sides. To characterize the shape of the responses (positive-then-negative waves or negative-then-positive waves), the latencies of the positive and negative peaks were compared. This analysis used a three-factor design with the factors of positive/negative spike, three head positions, and two sides of stimulation. *Post-hoc* testing with Bonferroni correction was conducted to identify significant differences ($p < 0.05$) between the levels of the different factors when appropriate. The effects size for the difference between the groups were calculated using Cohen's $d$, and were reported between brackets in results section.

**Table 1** **Response rate.** Percentage of participants displaying muscle response amplitudes exceeding three times the baseline standard deviation under different stimulation conditions.

| | HC-SC | HU-SC | HI-SC | HC-SI | HU-SI | HI-SI |
|---|---|---|---|---|---|---|
| **SCM RS1** | | | | | | |
| Positive | 12.5 | 18.75 | 56.25 | **93.75** | 68.75 | 31.25 |
| Negative | 81.25 | 68.75 | 37.5 | **6.25** | 31.25 | 50 |
| NoResponse | 6.25 | 12.5 | 6.25 | **0** | 0 | 18.75 |
| **SCM RS2** | | | | | | |
| Positive | 15 | *unmeasured* | 40 | **95** | *unmeasured* | 30 |
| Negative | 60 | *unmeasured* | 0 | **5** | *unmeasured* | 30 |
| NoResponse | 25 | *unmeasured* | 60 | **0** | *unmeasured* | 40 |
| **ES-C RS2** | | | | | | |
| Positive | 35 | *unmeasured* | 75 | **80** | *unmeasured* | 30 |
| Negative | 50 | *unmeasured* | 20 | **20** | *unmeasured* | 60 |
| NoResponse | 15 | *unmeasured* | 5 | **0** | *unmeasured* | 10 |
| **ES-L RS1** | | | | | | |
| Positive | 31.25 | 6.25 | 12.5 | 25 | 31.25 | 31.25 |
| Negative | 25 | 43.75 | 43.75 | 25 | 25 | 37.5 |
| NoResponse | 43.75 | 50 | 43.75 | 50 | 43.75 | 31.25 |
| **G RS1** | | | | | | |
| Positive | 37.5 | 25 | 25 | 12.5 | 18.75 | 31.25 |
| Negative | 25 | 43.75 | 37.5 | 25 | 37.5 | 18.75 |
| NoResponse | 37.5 | 31.25 | 37.5 | 62.5 | 43.75 | 50 |
| **TA RS1** | | | | | | |
| Positive | 6.25 | 25 | 25 | 31.25 | 31.25 | 18.75 |
| Negative | 25 | 6.25 | 12.5 | 0 | 12.5 | 18.75 |
| NoResponse | 68.75 | 68.75 | 62.5 | 68.75 | 56.25 | 62.5 |

## RESULTS

### Muscle response rate

The EMG vestibulospinal responses were found to be influenced by the type of muscle and the position of the subject (Table 1). We observed that responses with magnitude exceeding three times SD of the pre-stimulus control period could be present for all muscles in all conditions, but not for all participants. There was a proximo-distal distribution of responsiveness, with the SCM and ES-C muscles showing the highest response rates, with, in the best responding condition (head contralateral and stimulus ipsilateral to the muscle, HC-SI), 100% of the participants displaying a response. The more distal muscles, such as the gastrocnemius (G) and lumbar erector spinae (ES-L), showed lower response rates, with 68% for G (for stimulus ipsilateral and head-up, SI-HU) and 47% for TA (head contralateral and stimulus ipsilateral to the muscle, HC-SI). Overall, these findings suggest that the responsiveness of the muscles to vestibulospinal stimulation varies depending on the muscle type and the specific condition, with proximal muscles generally exhibiting stronger responses compared to distal muscles.

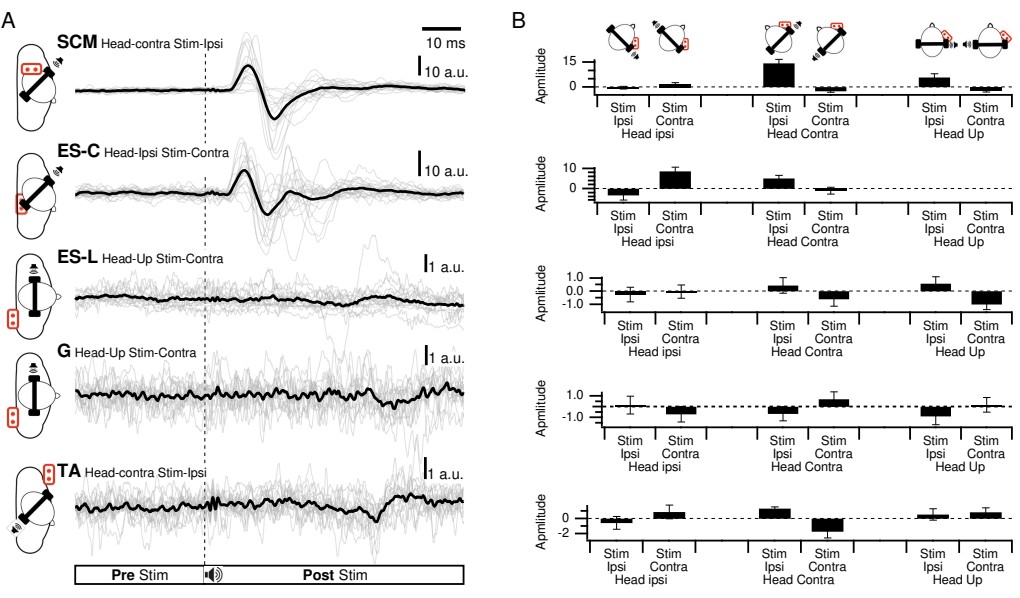

**Figure 2 Responses of all participants for each muscle.** (A) Responses of all participants in the best response condition for each muscle. Thin lines are the responses of each participant, thick line is the grand average. (B) Average amplitudes of the first response exceeding three times the standard deviation of the pre-stimulation period.

## VEMP waveform is dependent on participant's spatial orientation

The VEMP waveform was investigated using statistical analysis of peak polarity and latencies. For all participants, the SCM and ES-C muscles showed similar regular waveforms (Fig. 2), while for the ES-L, G and TA, the lower response to noise ratio resulted in significant inter-subject variability, making it difficult to establish reliable algorithmic characterizations.

In SCM, a clear waveform could be determined with a first positive peak in the head-up condition and for ipsilateral stimulation (HU-SI) in 69% of the participants. It was also positive when the stimulation was located in front of the subject (head contra-stimuli ipsilateral (HC-SI) and head ipsi - stimuli contralateral (HI-SC)), for 94% (95%[rs2]) and 56% (40%[rs2]) of the participants, respectively (Fig. 3). The first peak magnitude was significantly influenced by head position and stimulus laterality ($F^{rs1}(2,20) = 21.2$, $p < 0.001$; $F^{rs2}(1,4) = 24$, $p = 0.008$) and were 5.8, 14.3 (19.9[rs2]), and 1.9 (10.8[rs2]) a.u. (arbitrary unit) for the HU-SI, HC-SI and HI-SC conditions, respectively [$d_{HU-SI/HC-SI}^{rs1} = 0.94$; $d_{HC-SI/HI-SC}^{rs1} = 1.96$; $d_{HU-SI/HI-SC}^{rs1} = 0.66$; $d_{HC-SI/HI-SI}^{rs2} = 0.86$]. The first peak latencies were not significantly affected by the head and stimulus position ($F^{rs1}(2,30) = .3$, $p = 0.77$; $F^{rs2}(1,4) = .9$, $p^{rs2} = 0.40$), with an averaged value of 17 ms[rs1] ($SD^{rs1} = 8$) (16 ms[rs2] $SD^{rs2} = 2$). In these three responsive conditions, a second negative peak occurred after 26 ms (SD =1ms), with magnitude of 7.6, 17.5 (18.2[rs2]), and 4.4 (13.5[rs2]) a.u. for HU-SI, HC-SI and HI-SC respectively ($F^{rs1}(2,6) = 17$, $p < 0.001$ [$d_{HU-SI/HC-SI}^{rs1} = 0.91$; $d_{HC-SI/HI-SC}^{rs1} = 1.86$; $d_{HU-SI/HI-SC}^{rs1} = 0.49$]; $F^{rs2}(1,2) = 389$, $p = 0.003$ [$d_{HC-SI/HI-SI}^{rs2} = 0.43$]). In the three remaining conditions (contralateral stimuli with head up, or back

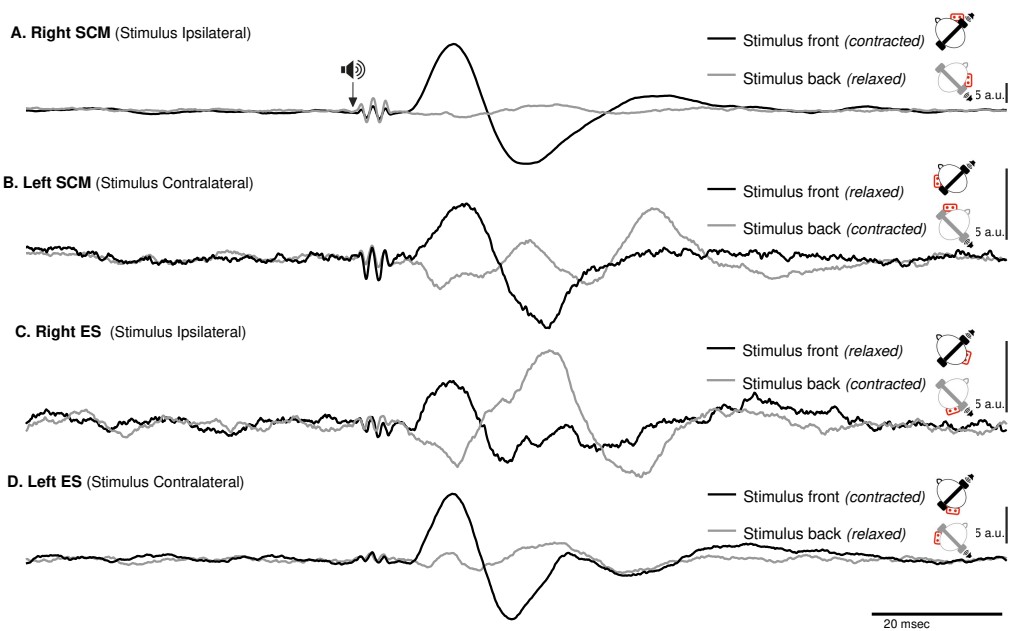

**Figure 3** **Spatial orientation of VEMP (SCM and ES-C).** Vestibular evoked myogenic potentials (VEMPs), averaged across all subjects, from the sternocleidomastoid (SCM) and ipsilateral erector spinae (ES-C) muscles on both the right and left sides. Tone bursts were consistently delivered to the participants' right ear (ipsilateral for the right-side muscle and contralateral for the left-side muscle). In all muscles, the only variation between the black and grey curves was the reversal of the head position, which resulted in VEMPs with reversed polarities. For all muscles, irrespective of muscle side or whether they were in a stretched or relaxed state, VEMP polarity depended on the spatial origin of the stimulus relative to the trunk (front or back).

stimuli) responses were weaker, with a negative peak first occurring at 19 ms (SD = 4; 20 ms[rs2] SD[rs2] = 9) for 69% (HU-SC), 82% (60%[rs2])(HC-SC) and 50% (30% [rs2]) (HI-SI) of the participants, with magnitudes of −2.3, −2.5 (−4.1[rs2]), and −0.3 (−0.5[rs2]) respectively. A positive peak followed, at 28 ms (27 ms[rs2]) (SD = 4, SD[rs2] = 8), not significantly different between the various conditions, with magnitude of 0.6, 2.8 (6.4[rs2]) and 1 (3.4[rs2]).

ES-C also exhibited a net standard waveform, with positive then negative peaks when the stimuli originated from the front, and negative then positive peaks when the stimuli originated from the back (Fig. 3). When the stimuli were front located, a first positive peak was detected in 80% (HC-SI) and 75% (HI-SC). The magnitude was significantly influenced by head position and stimulus laterality ($F_{(1,15)}$ =16.7, $p < .001$), but post hoc analysis revealed no difference between HC-SI and HI-SC conditions (5.1 and 8.6 a.u. respectively). The latencies of this first detected peak was 17 ms (SD = 4), not significantly different between HC-SI and HI-SC, despite a significant head x stimulus interaction ($F_{(1,15)}$ = 12.9, $p = .003$). A second negative peak occurred after 27 ms (SD = 4), with magnitudes of −5.9 and −9.6 a.u. (no significant difference, NSD). In the two other conditions, with stimuli at the back, the first peak was negative for 67% (HI-SI) and 59% (HC-SC) of participants. It occurred after an average latency of 19 ms (SD = 8) with magnitudes of −3.3 and −1 (NSD) for the HI-SI and HC-SC conditions, respectively. Following the

negative peak, the second peak was positive for both conditions. The magnitudes of the second peak were 6.5 and 2.3 (NSD) for the HI-SI and HC-SC conditions, respectively. It occurred after an average latency of 27 ms (SD = 8).

For the magnitude of the first peak of both ES-C and SCM muscles, an interaction between stimulus laterality and head position was reported from rs2 ($F^{rs2}(1,15) = 16.7$, $p < 0.001$ for ES-C, and $F^{rs2}(1,4) = 24$, $p = 0.008$ for SCM). This interaction effect can be attributed to the significant difference between conditions with a positive-negative waveform (front stimuli) and conditions with a negative-positive waveform (back stimuli). Specifically, the conditions with stimuli coming from the front of the subject (HC-SI and HI-SC) showed one pattern of waveform (positive-negative, Fig. 3 upper panel), while the conditions with stimuli coming from the back of the subject (HC-SC and HI-SI) exhibited a different waveform pattern (negative-positive, Fig. 3 lower panel). Post hoc analysis further revealed that all conditions with back stimuli had first peaks with magnitudes significantly different from those in the conditions with front stimuli. This indicates that the magnitude of the first peak in response to back stimuli differed significantly from that in response to front stimuli for both the ES-C and SCM muscles.

When examining, the first peak that exceed the baseline by $3 \times$ SD in ES-L, it was not possible to define a common standard waveform. There was no significant effect of stimulus side and/or head orientation on the first peak amplitude (stimulus side $F(1,1) = 0.05$, $p = 0.85$; head orientation $F(2,2) = 1.73$, $p = 0.37$; interaction $F(2,2) = 2.3$, $p = 0.3$), and the averaged peak magnitude remained under 1 a.u. In the best responding condition (HI-SI), the participants exhibited a first peak that was positive (31%), or negative (38%) while 31% of participants did not exhibit any response. Furthermore, peak latencies displayed an important interindividual variability (20 ms, SD = 13).

Similarly, the low response rate in the G and TA muscles did not allow repeated measures ANOVA on the first-peak magnitudes and latencies. In the best responding condition for the G muscle (HU-SC), the first peak was positive in 44% of participants, negative in 25% and absent in 31% of participants. Latency values were highly variable (17 ms, SD = 10). With regards to TA in the best HU-SI responding condition, the first peak was positive in 31%, negative in 13% and absent in 56% of participants.

## DISCUSSION

In our study, the primary objective was to identify the optimal conditions within the same experimental setting for detecting tone burst-elicited VEMPs in various muscles. (ES-C, ES-L, G, TA) taking the SCM muscle responses as a reference. We found that VEMPs were consistently detected in the SCM and ES-C muscles under optimal conditions, but not in the G, ES-L and TA muscles. Furthermore, we investigated how these vestibulospinal reflexes were influenced by the head position relative to the cervical spine.

### VEMP responses in the various muscle groups

When considering the primary role of the trunk in maintaining posture (*Guillaud et al., 2020b*), it is important to investigate whether the tone burst method allows for studying the vestibulospinal reflex in axial muscles. Previous studies (*Ardic & Daniel Latt, 2000*; *Iles, Ali*

*& Savic, 2004*) have shown a bilateral paraspinal response during GVS. To the best of our knowledge, this study would be the first to observe VEMPs in the axial muscles of the trunk. We validated our methodology by recording the vestibulospinal reflex in the SCM muscles and obtained results similar to previous studies (*Rosengren et al., 2019*). VEMP recordings in distal muscles confirm the presence of vestibular projections throughout the entire axial muscle chain and the appendicular system. We observed, consistent with previous reports (*Rudisill & Hain, 2008*; *Guillaud et al., 2020a*), that the electromyogram (EMG) response latencies increased from rostral to the most distal recorded muscles. These latencies remained short, suggesting a fast integration process between the vestibular nuclei and the motoneurons. However in this study, only the more rostral muscles (SCM and ES-C) exhibited reproducible VEMP patterns. More caudal muscles (ES-L, TA, G) presented myopotential waveforms that change between subjects. The reason for heterogeneous responses in distal muscles is currently unclear. It may be related to electrode placement, despite taking precautions to ensure proper positioning. An higher variability in distal muscle activity could also lead to this heterogeneity. While we lacked measures of center of pressure or kinematics, it is reasonable to assume that the slight ahead lean of our participant involves constant postural oscillations, as during stable upright (*Zatsiorsky & Duarte, 2000*). Given the influence of the subject's estimated center of gravity position on the amplitude of GVS-evoked muscle responses (*Son, Blouin & Inglis, 2008*), our "challenging posture" likely heightened variability in ankle and lower back muscle responses.

Vestibulo-spinal reflexes elicited by GVS at muscle limb level show a more reproducible and stable, head-position dependent pattern than what is observed in the present report for sound-evoked reflexes. Therefore, the difference between neck and limb muscles observed in the present study is unlikely to be a general characteristics of the VS reflexes, but it seems to depend on the technique utilized for eliciting these reflexes. While tone-burst stimulation is believed to be confined to the saccule, other stimulation techniques (specifically, GVS) have been demonstrated to stimulate both otolith and canal afferents (*e.g.*, see *Forbes et al., 2023*). Thus, compared to tone-bursts, GVS is more likely to activate both MVST and LVST, the latter of which activates the musculature of the lower body. Therefore, GVS could be more effective at activating postural muscles in the lower body.

These different responses could be also attributed to the distinct roles of recorded muscles in maintaining balance, as well as the organization of the vestibulospinal pathways. Indeed, the modulation of EMG responses in appendicular muscles, compared to axial muscles, is more related to body contact and support from the surroundings rather than postural attitude. Therefore, the gating of vestibular input, which relies on the integration of various sensory information (visual, proprioceptive, and vestibular), would not be the same for different muscles. The vestibular drive to the appendicular muscles exhibits greater flexibility, as it can be suppressed by somatosensory inputs such as tactile contact or during locomotion (*Ali, Rowen & Iles, 2003*; *Fitzpatrick & Day, 2004*). Some of the differences observed between axial and appendicular muscles may be related to the existence of two pathways (*Guillaud et al., 2020a*). The first pathway is the lateral vestibulospinal tract (LVST), which extends throughout the entire length of the spinal cord and can activate both axial and hindlimb muscles (*Graf et al., 1997*; *Shinoda et al., 2006*; *Kasumacic, Glover*

*& Perreault, 2010*; *Kasumacic et al., 2015*). The second pathway is the medial vestibulospinal tract, which primarily influences musculature in the upper body (*Akaike, 1983*; *Perlmutter et al., 1998*). *Rudisill & Hain (2008)* concluded that the low response rate observed in the G muscle (62.5% of participants) does not exhibit sufficient reproducibility to validate its use as a clinical test. The same conclusion can be drawn here for the ES-L and TA muscles.

## Spatial dependency of vestibulospinal reflex integration

In addition to investigating the detection of VEMPs in various muscles, we specifically focused on exploring how the vestibulospinal reflexes were influenced by head position relative to the cervical spine. We recognized the significant role of head position in modulating vestibular responses and aimed to gain a deeper understanding of how this factor affected the observed VEMPs in the selected muscles. By conducting our study using a consistent experimental framework, we were able to directly compare and evaluate the VEMPs elicited in different muscles under various conditions. We observed similar results on the SCM and ES-C muscle, with VEMPs exhibiting the same polarity (Fig. 3).

For a unique side of stimulation, it was possible to record similar responses in both the ipsilateral SCM and contralateral ES-C muscles (Fig. 3), indicating a bilateral response. This finding suggests that the vestibulospinal pathway involved in these reflexes is not strictly ipsilateral (*Ashford et al., 2016*) but also involves crossing of neural signals at the midline. This is in line with previous studies that have suggested the existence of commissural connections within the vestibulospinal system (*Pompeiano, Mergner & Corvaja, 1979*). This effect is, however, not observed with GVS (*Forbes et al., 2018*), and this discrepancy may be attributed to differences in the nature of the stimulus. GVS exerts complex effects (for a review, see *Fitzpatrick & Day, 2004*), resulting from an electric flow between the cathode and anodes, each positioned on one ear. Therefore, the overall effect arises from a combination of cathodal and anodal stimulation, while tone-burst only affects one ear (as it is unlikely that, due to the focal mechanical stimulation delivered, there was a crossed activation of the other inner ear).

For a single side of stimulation in a given head position, all four muscles exhibited responses of similar polarity at the same latencies (Fig. 3). When stimuli were applied from the front, the polarity was positive-negative, and the responses were more prominent compared to when applied from behind. This held true regardless of whether the muscle (ES or SCM) was contracted or relaxed, although the signal-to-noise ratio remained better when the muscle was contracted. Additionally, for a single side of stimulation and each muscle, we observed a change in polarity in the VEMPs when head orientation was altered (Fig. 3). This finding indicates that the polarity of the VEMPs is not solely determined by the recording site, such as the placement of the active electrode relative to the motor point. Other factors, such as the orientation of the head, can influence the polarity of the VEMPs. When the head is rotated, the orientation of the vestibular receptors is modified relative to the rest of the body. Assuming similar mechanisms, these results are consistent with those observed using GVS, where lateral disturbance illusion become antero-posterior with head turned (*Pastor, Day & Marsden, 1993*; *Fitzpatrick & Day, 2004*). Although we did not measure the postural compensations induced by acoustic stimulation, it constituted a

sufficient perturbation to elicit a functional vestibular response in the form of VEMPs. The large response recorded for ipsilateral stimulation could be interpreted as the detection of a horizontal translation, or tilt of the head away from the stimulation. However, this response to ipsilateral stimulation was bilateral and appeared simultaneously and with the same polarity for these two groups of posterior and anterior neck muscles, although it remained greater on the contracted SCM and ES-C muscles. This suggests that the VEMP stimulation used in this study corresponds to a more complex head displacement than a simple unidirectional displacement. It was unlikely, however, that a unilateral otholitic activation, presumably sacular (*Welgampola & Colebatch, 2005*), could be interpreted as a simple displacement. Despite the likely inconsistency in stimulation, our results still demonstrate a spatial integration of the stimuli. These observations highlight the complexity of the vestibulospinal reflex pathways and the factors that can influence their responses.

Regarding the exploration of vestibulospinal pathways, our study reveals that the tone-burst method has some limitations, which, however, are not specific to tone-burst alone. While GVS allows for a more sustained activation of vestibulospinal pathways in axial and leg muscles (*Guillaud et al., 2020a*), the participants' experiences indicate that it causes more discomfort than the tone-burst method. This is particularly true for fragile patients, such as those undergoing vestibular neurotomy surgery, whose vestibular function one aim to explore in the postoperative period. Thus, these two methods are not antagonistic but complementary.

In conclusion, our study provides valuable insights into the complex interplay between head position, muscle activation, and vestibulospinal reflexes. Our results suggest axial and distal VEMP in ES-L, G and TA, but the high inter-individual variability of responses makes it unsuitable for clinical use. Overall, our study contributes to a better understanding of the vestibular responses in different muscles and their modulation by head position. It may have implications for future research and clinical applications related to vestibular function assessment and rehabilitation.

### Funding
This work was supported by CNES (Centre National d'Etudes Spatiales, France, DAJ/AR/LF-2020.0033722). The funders had no role in study design, data collection and analysis, decision to publish, or preparation of the manuscript.

### Grant Disclosures
The following grant information was disclosed by the authors:
CNES (Centre National d'Etudes Spatiales, France, DAJ/AR/LF-2020.0033722).

### Competing Interests
The authors declare there are no competing interests.

![PeerJ]

## Author Contributions

- Vincent Jecko performed the experiments, analyzed the data, prepared figures and/or tables, authored or reviewed drafts of the article, and approved the final draft.
- Léa Garcia conceived and designed the experiments, performed the experiments, analyzed the data, authored or reviewed drafts of the article, and approved the final draft.
- Emilie Doat performed the experiments, analyzed the data, prepared figures and/or tables, and approved the final draft.
- Vincent Leconte analyzed the data, prepared figures and/or tables, and approved the final draft.
- Dominique Liguoro conceived and designed the experiments, authored or reviewed drafts of the article, and approved the final draft.
- Jean-René Cazalets conceived and designed the experiments, prepared figures and/or tables, authored or reviewed drafts of the article, and approved the final draft.
- Etienne Guillaud conceived and designed the experiments, analyzed the data, prepared figures and/or tables, authored or reviewed drafts of the article, and approved the final draft.

## Ethics

The following information was supplied relating to ethical approvals (i.e., approving body and any reference numbers):

The experiments conducted in this study were approved by the ethical research committee (French Comité de Protection des Personnes Sud-Méditerranée I, ID RCB 2019-A02985-52).

## Data Availability

The raw data and code are available in the Supplemental Files.

## Supplemental Information

Supplemental information for this article can be found online at http://dx.doi.org/10.7717/peerj.17056#supplemental-information.

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
