# Peer review of "Vestibulospinal reflexes elicited with a tone burst method are dependent on spatial orientation"

_PeerJ, doi:10.7717/peerj.17056_

## Round 0.1 · original submission · Minor Revisions

The reviewers were generally enthusiastic and found no major problems in the design and statistical approaches. The lack of a clear idea flow that establishes the significance and impact of the study is the main current shortcoming in this submission. Please clarify the arc of the result description and supplement it with the general conclusions that extend the findings in the study to the knowledge in the field.

·

Basic reporting

This paper is an interesting contribution to the scarce literature on vestibulospinal pathway physiology.
The text is well written even when the discussion could be shorter.
The article structure is correct.
Some hypotheses are opinioned.

Experimental design

The research is within the Aim and Scope of the Journal
The author (line 107-117) affirm that the sound stimulus characteristics described earlier were determined to be optimal for eliciting VEMP responses with maximum amplitude while using the lowest possible stimulus intensity. These characteristics were chosen to ensure reliable and consistent responses without inducing learning or fatigue.
Where does this statement come from?
They choose to test with Head Rotation to both sides, why? To test a VS effect, especially mediated by MVST to tilt the head has more sense.

Validity of the findings

Even when the article provides interesting data and statical analysis and data are correct, some conclusions like (lines 359-361): "We have demonstrated the bilateral nature of the vestibulospinal pathway and the influence of head orientation on the polarity of vestibular evoked myogenic potentials (VEMPs). These findings underscore the involvement of multiple neural mechanisms and integration processes in generating VEMPs." these are not supported by the data and have to be revised.

·

Basic reporting

.

Experimental design

The goal is clear and the result is sound. The topic is of interest in the field of vestibular physiology and its clinical applications

Validity of the findings

The description of the results should be improved. Many point of the discussion have to be clarified.

Additional comments

Here a detailed list of the comments
General consideration
The goal of the paper is to verify whether sound-evoked Vs reflexes are tuned by the position of the head with respect to the body. This appear to be the case. The result is sound and there are no major methodological or statistical problems. The results are very difficult to follow, partly for the inherited complexity of the topic, partly for the lack of a comprehensive picture of the different head positions and stimulated sites referred to a single muscle. In this way the reader could associated the muscle responses to the different conditions that are based on 1) the position of the of the head with respect to the muscle, 2) the position of the stimulus with respect to the muscle and 3) the position of the stimulus with respect to the subject (front, back). In this respect figure 2 could be modified by inserting above the columns relative to muscle activation the figures displaying the subject’s head orientation and the stimulus source. Probably, for allowing a better definition of the head/stimulus condition 2A and 2B can be split in two separate figures. Finally, several part of the discussion should be clarified (see specific comments).

Specific comments
Abstract
Lines 23-24.“This study aims to investigate the effect of tone burst stimulation on vestibulospinal reflexes”. Since tone burst stimulation elicits VS reflexes I would suggest to say “to investigate vestibulospinal activation elicited by tone burst stimulation”
Introduction
Line 42. If there are paper addressing a vestibular contribution to these symptoms, they should be quoted.
Line 45. If this technique is gold standard, there should be an appropriate reference to report here, may be choosing among those of lines 48-49, if it is the case.
Line 53. Vestibulocollic
Lines 65-67. May be it would be better to talk about integration of vestibular, visual, proprioceptive and somatosensory information, or else of modulation of vestibular responses by visual, proprioceptive and somatosensory information.
Line 74. I would suggest to use the expression: However, there is currently a lack of reports on such interaction when the VS reflexes are elicited under acoustic stimulation.
Methods
Lines 102-103. The stimulus profile should be shown by a figure. The term trapezoidal sinusoid is unclear and not commonly used.
Lines 152-154. The phrase is unclear: the contact with the floor is always maintained through the feet. This does not reduce the proprioceptive/somatosensory input. Or am I missing something?
Lines 155-160. Is there any important difference between rs1 and rs2, a part the recording of cervical erector spinae and the lack of head up position testing in rs2? I could not catch the rationale of these two different sessions by reading the methods.
Lines 181-183. I would suggest to make the following correction in order to improve the clarity. “The stimulations (S) and head orientations (H) were classified as ipsilateral (SI and HI) or contralateral (SC and HC) to each recorded muscle in order to analyze the measurements of both right and left muscles of the participants”
Line 183. “These responses were considered symmetrical” the phrase is unclear. Responses are similar on both sides (symmetric) or are not (asymmetric). This is a fact that has to be simply observed, not assumed. Probably I am missing something.
Line 186. Please detail (at least) the head positions (HU, HI, HC): the reader has probably forgot the HU position
Line 190-191. Why did the authors used a one way design instead of a repeated measures Anova with stimulation and position as within subject factors?
Results
Line 218. How the Authors explain the big difference in the response amplitude observed between the rs1 and rs2 in HI-SC?
Lines 230-232. Please indicate in the figure not only the position of the stimulus, but also the direction of head turning with respect to the recorded muscle: the term front/back are not enough for an easy comprehension of the data.
Discussion
Lines 296-307. VS reflexes elicited by GVS at muscle limb level show a more reproducible and stable, head-position dependent pattern than what observed in the present report for sound-evoked reflexes. So, the difference between neck and limb muscles observed in the present study is unlikely to be a general characteristics of the VS reflexes, but it seems to depend on the technique utilized for eliciting these reflexes.
Lines 326-329. Why a stronger response following front stimulation is indicative of a synergistic vestibular/proprioceptive interaction? Because of the different position of the head in the two conditions? Please clarify.
Lines 330-331. This is at variance with the effects of the galvanic stimulation of the labyrinth ( Forbes et al., Frontiers Neurol 2018). The authors should discuss this difference between the two methods. Can a bilateral activation of the labyrinth by sound be excluded with certainity?
Lines 341-354. If the message is that sound generates a bilateral activation of the two labyrinths, more similar to the effects of a forward displacement than to a lateral displacement or to a tilt, it should be expressed more clearly. In this respect, there are data in the literature relative to bilateral cathodic stimulation of the labyrinth? If this is so, are they similar to the results obtained in the present report?
Lines 345-347. A response to tilt-translation generally induces responses of opposite sign on the two sides. In this sense,
Line 350-356. The paragraph should be clarified. Although it is very questionable to compare the effect of sound stimulation to those of forward (symmetric activation of the two labyrinths) or laterally directed (opposite patterns of activation of the two labyrinths) displacements, the authors have to report how these displacements affect muscle activity in the neck muscles. This may allow a better understanding of the paragraph.

Reviewer 3 ·

Basic reporting

No comment

Experimental design

I do have some concerns here:

- I am curious why the authors used a postural lean for the leg muscle testing? Perhaps just having the participants attempt to “stand straight” would have been better and less prone to variability in the background muscle activation? Did the authors confirm that the participants were able to consistently able to keep themselves at 15 deg forward lean? This seems hard to do without having them on a force plate or measuring ankle angle or centre of gravity. Perhaps this explains the heterogeneous responses seen in these muscles, as the background muscle activity was varying widely even within a given trial. Also, leaning forward will cause TA to essentially shut off, which is not ideal for attempting to observe a reflex response.

- A 5.2 Hz rate seems quite high to me. The authors mention that this did not induce learning or fatigue, but how was this confirmed? If you were to compare responses occurring on the first half of each trial to those during the last half of each trial, is there really no change in the evoked response magnitudes? It would seem like a longer ISI would be preferrable to not cause response adaptation.

- Results section – I believe it is now standard to report effect sizes?

Validity of the findings

- The authors mention that tone-bursts are currently the clinical gold-standard, however, based on their own results, it appears that it is unable to generate responses in the axial and appendicular musculature. These lower body muscles are critical for postural control, thus, it would seem that tone-bursts are severely limited in what they can tell the clinician about vestibular function. GVS is a well known alternative approach that can generate myogenic responses across the body, and I would argue from extensive experience that it is entirely safe and non-invasive enough to be of concern to clinicians. Do the authors not think that supplementing clinical diagnoses with GVS testing is worthwhile, especially in the case where deficits may be limited to the lower extremities or where postural control is being affected? I would like to see a more nuanced discussion of this topic in the discussion/conclusion section.

- The authors state that “The reason for heterogeneous responses in the distal muscles is unclear”. However, one undiscussed explanation for this finding is that, while tone-burst stimulation is thought to be limited to the saccule, other stimulation techniques (namely, GVS) have been shown to stimulate both otolith and canal afferents (e.g, see Kwan et al 2023 - The Neural Basis for Biased Behavioral Responses Evoked by Galvanic Vestibular Stimulation in Primates. J Neurosci). Thus, compared to tone-bursts, GVS is more likely to activate both MVST and LVST, the latter of which activates the lower body musculature. This could likely explain why GVS is better at activating postural muscles of the lower body. I would like to see this discussed in the manuscript.

Additional comments

- Line 166 – I assume ‘ECG’ is supposed to be ‘EMG?

---

## Round 0.2 · accepted · Accept

Thank you for the high-quality revision.

·

Basic reporting

The authors have satisfactorily answered to the criticisms and the revised version has been improved with respect to the original. The paper can be accepted in the present form

Experimental design

OK

Validity of the findings

OK

Reviewer 3 ·

Basic reporting

I have no concerns regarding basic reporting in this manuscript.

Experimental design

The experimental design is sound and the authors have cleared up my previous concerns in their response letter.

Validity of the findings

The findings are valid, and I believe the additions made by the authors in this round of reviews helped further improve the readers ability to appreciate this.

Additional comments

I have no further comments and would be happy to have this article published as is.